# Knowledge and Attitude of Medical Students and Doctors Towards Artificial Intelligence: A study of University of Ilorin

February 28, 2023

### Abstract

This study assesses the knowledge and attitudes of medical students and doctors in University of Ilorin toward Artificial Intelligence (AI) in medical education. It involved a cross-sectional study using an online survey consisting of close-ended questions. The survey targeted medical students at all medical levels and doctors in their postgraduate training. total of 481 medical students and doctors responded. When assessing AI knowledge sources (86.5%) got their information from the media as compared to (13.5%) from medical school curriculum. However, students who learned the basics of AI while being in medical school were more knowledge about AI than their peers who did not and were more interested in the applications of AI in healthcare. The advancements in AI affected the choice of specialty of around a quarter of the students (26.8%). Finally, less than a quarter of students (22.1%) want to be assessed by AI, even though about half (57.7%) reported that assessment by AI is more objective

## 1  Introduction

Artificial intelligence (AI) has received remarkable attention for a very long time and is sometimes referred to as the fourth industrial revolution (Abid et al, 2019). Over the past few decades, artificial intelligence (AI) has grown in popularity, and its application in medicine is expanding on a global scale (Ahmed et al, 2022).

Despite the widespread use of AI in healthcare in developed countries, developing countries in Africa are still behind in the use, research, education and implementation of AI in healthcare (Owoyemi et al., 2020). The majority of AI application in the healthcare industry is witnessed in radiology (Waymel et al., 2019) using imaging techniques of AI system, computer vision to identify and understand different malignant entities (Hosny et al., 2018). It is also being employed in gastroenterology in the context of scoping to detect pathological lesions (Alagappan et al., 2018).

Although it is widely acknowledged that AI will play a significant role in medicine, it is still unclear how this will affect medical students and their future in Nigeria. The study of AI is not frequently discussed in medical school and not addressed enough. Current accreditation standards do not emphasize AI, medical schools are already struggling with a dense curriculum and are frequently asked to add new topics and areas of study, and they lack faculty with the subject matter expertise and technical know-how to teach this topic are a few of the potential explanations given in a study (Kolachama et al., 2018).

## 2    Materials and Method

During a six week period, we distributed a google form containing closed-ended questions to medical students and physicians in University of Ilorin using social media apps e.g. Whatsapp. This research was a cross-sectional study. Responses were anonymous without any identifying data, and only the principal investigator had access to the data. A convenience sampling technique was used to pick the sample population.

## 3    Results

The mean score of Knowledge of AI was 1.82 ± 1.83. Regarding Knowledge of AI, individuals were questioned about the basic concept of AI, its subtypes, i.e., machine learning (ML) and deep learning (DL), and its applications. It was observed that 336 (70%) had a basic concept of AI, but only 162 (34.7%) had Knowledge about ML and DL. The mean score of attitudes toward AI was 6.03 ± 2.16. On the necessity of inclusion of AI in the medical field, 219 (45.7%) individuals strongly agreed. Regarding the opinion that AI aids practitioners in early diagnosis and assessment of disease severity, 177 (37%) strongly agreed. The idea that AI can replace the physician in the future, 127 (26%) strongly agreed,

## 4    Conclusion

Although most physicians and medical students do not sufficiently understand AI and its significance in the medical field, they have favorable views regarding using AI in the medical field. Nigerian medical authorities and international organizations should suggest including artificial intelligence in the medical field, particularly when training residents and fellowship physicians.

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
