# OpenReview forum: "Knowledge and Attitude of Medical Students and Doctors towards Artificial Intelligence: A study of University of Ilorin"
_ICLR.cc/2023/TinyPapers — Submitted to Tiny Papers @ ICLR 2023_

### Official Review · Reviewer_6DW1 · 2023-03-27

**Confidence:** 4

**Summary Of Contributions:**

The authors perform a cross-sectional study using an online survey to assess the perception of AI from medical students and practitioners in Africa.

**Rating:**

Needs Clarification (NC): a submission which does not meet the reviewing criteria and needs clarification for its described problem or solution

**Strengths And Weaknesses:**

Strengths:
- This is an important study. There's usually a resistance towards applying AI to medical fields and it's important for us to understand what are the basic requirements the medical community needs to trust these softwares.

Weaknesses:
- There are some grammar and spelling problems throughout the text. Would be nice to use a free software to help with this.
- The survey methodology could be explained in more detail.

**Suggested Changes:**

- Please use the ICLR Tiny Paper track template, instead of the raw overleaf template. I recommended the rating of NC basically because it does not comply to the ICLR tiny paper formatting requirements. Otherwise, I'd be comfortable giving this a Great Start review!
- Were there similar studies performed you could compare to or base yourselves on? Maybe even from outside of Africa. It'd make your paper more impactful if you replicated questions and methodologies done in prior work. You could eventually compare how is Nigeria's perception of AI versus other countries and regions in the world.
- I'd like to have more insights about the questions, possible answers, maximum scores from the form and methodology of results collection/aggregation. The paper would also benefit from some images and tables showing results to the main topics of interest.

---

### Official Review · Reviewer_oMHN · 2023-04-01

**Confidence:** 3

**Summary Of Contributions:**

The paper assesses the awareness and knowledge of artificial intelligence in medical student community in Nigeria

**Rating:**

Needs Clarification (NC): a submission which does not meet the reviewing criteria and needs clarification for its described problem or solution

**Strengths And Weaknesses:**

The contribution of this paper is sharing results of a survey done  on medical students of Nigeria to assess their familiarity/knowledge on artificial intellligence. It concludes that AI should be included in the curriculum.

While this shares useful insights, this might not be a relevant conference for this work. The conference focuses on representation learning in multiple domains and this work does not associate with it.

**Suggested Changes:**

Minor mistakes in abstract:

“Total of 481…” : uppercase the word total

“Were more knowledge..” : grammatical error

“Objective”: add a period.

---

### Comment · Area_Chair_tbgF · 2023-06-06
**Archival Criterion Check**

Since the authors didn't provide a revised version, it fails to meet the archival criterion.

---

### Meta-Review · Area_Chair_tbgF · 2023-04-08

**Recommendation:** Invite to revise
**Confidence:** 4

**Metareview:**

The authors investigate the familiarity of medical students in Africa with artificial intelligence and conclude that AI should be included in the curriculum. While the topic is interesting for the broader AI community, it falls somewhat disconnected from the ICML Tiny Paper's vision. The summary of pros and cons from the reviewers' comments is as follows.

Strength:

1. The paper shares useful insights from a survey conducted among Nigerian medical students to assess their familiarity with AI.
2. It highlights the importance of understanding the medical community's needs and trust in AI software.

Weakness:

1. The paper may not be relevant to the conference, which focuses on representation learning in multiple domains.
2. There are grammar and spelling issues throughout the text.
3. The survey methodology could be explained in more detail.

**Summary:**

The paper evaluates AI awareness among Nigerian medical students and practitioners through a cross-sectional online survey assessing their perception of AI in Africa.

**Comments And Feedback To The Authors:**

I believe the authors are investigating an interesting topic, exploring how people's exposure to AI affects their understanding, which could have a significant impact on future education strategies to embrace AI. However, as both reviewers mentioned, the paper falls short in several criteria, such as clarity and reproducibility. I encourage you to work more on the problem and wish you all the best in your future academic endeavors.

**Reason For Not Giving A Higher Recommendation:**

The paper fails short in reproduction criterion. And as mentioned by both reviewers, current form of study seems to be irrelevant to the conference vision. To improve on the current manuscript, I encourage authors to add similar studies that can compared to, clean up the grammar errors, use the correct format, and more detail on the method and the analysis.

**Reason For Not Giving A Lower Recommendation:**

N/A

---

### Decision · Program_Chairs · 2023-04-08

No revision received; not invited to archive